# Internal Decay in Landscape Oaks (*Quercus* spp.): Incidence, Severity, Explanatory Variables, and Estimates of Strength Loss

**Nicholas J. Brazee** [1,*] and **Daniel C. Burcham** [2]

[1] Center for Agriculture, Food and the Environment, University of Massachusetts, Amherst, MA 01003, USA

[2] Department of Horticulture and Landscape Architecture, Colorado State University, Fort Collins, CO 80523, USA; daniel.burcham@colostate.edu

[*] Correspondence: nbrazee@umass.edu

**Abstract:** As trees age, internal decay and the risk of stem failure become important management issues for arborists. To characterize the incidence and severity of internal decay in landscape oaks, 323 pairs of sonic and electrical resistance tomograms were generated from 186 trees, representing five species: *Q. alba*, *Q. bicolor*, *Q. palustris*, *Q. rubra*, and *Q. velutina*. Overall, 135 (73%) oaks had detectable decay. When intermediate sonic velocities were included, the mean area of decay ($A_D$) was 41% with a mean strength loss ($Z_{LOSS}$) of 35%. Among all oaks, *Q. rubra* had the highest frequency of decay symptoms and signs of a pathogen. Binomial regression showed that diameter, symptoms, and oak species were the best predictors of decay incidence, and beta regression showed that diameter, scanning height, and species were the best predictors of decay severity. *Quercus alba* had the highest mean $A_D$ while *Q. bicolor* or *Q. palustris* had significantly less decay, depending on tomogram interpretation, across all modeled conditions. Despite considerable variability, the empirical model of decay incidence and severity fit to tomography measurements can inform decay assessments of landscape oaks, but the detailed tomograms allowed more precise strength loss estimates, especially for offset decay columns.

**Keywords:** sonic tomography; electrical resistance tomography; butt rot; wood-rotting fungi; fungal pathogens; urban forestry

## 1. Introduction

Across northeastern North America, oaks (*Quercus* spp.) are some of the most abundant, valuable, and largest trees in developed (i.e., urban and suburban) landscapes. In the Frank A. Waugh arboretum on the University of Massachusetts campus (Amherst, MA, USA), oaks make up 9% of all trees and 22% of total tree value, based on the Council of Tree and Landscape Appraisers (CTLA) calculations [1]. Additionally, oaks represent 28% of all trees with a diameter at breast height (DBH) ≥76.2 cm. Their adaptability and tolerance of urban stresses have made oaks a desirable choice for a variety of landscape sites [2].

However, as these trees continue to age, internal decay caused by wood-decaying fungal pathogens is a common concern. Internal decay in the roots and trunk decreases the structural stability of infected trees, making them more susceptible to uprooting and stem failure [3], especially under loading from strong winds [4]. Decay in the roots and lower trunk of oaks is often initiated through wounds [5]. In developed landscapes, oaks are subjected to a range of mechanical injuries, such as construction damage to roots [6] and wounding from lawn care equipment [7]. Furthermore, the majority of wood-decaying fungi that attack oaks colonize the heartwood after establishment [3,8]. As a result, advanced infections can develop without any visible symptoms or impairment of the vascular tissue. While oaks have strong, durable wood, they are susceptible to infection and decay from a wide array of fungal pathogens [8]. Previous studies on the damage caused by decay pathogens of oak have primarily focused on timber losses in forest settings [9–11].

These studies have helped to identify the most common and destructive wood-decaying pathogens, many of which occur across urban and suburban landscapes [3,12].

Studies on the incidence and severity of internal decay in landscape trees are lacking. This is a significant knowledge gap, given the importance of fungal decay in tree failure [13] and the resulting legal ramifications [14]. Yet, some studies provide insights into decay incidence. A review of data from the California Tree Failure Report Program (https://ucanr.edu/sites/treefail/) (accessed on 2 November 2022) found that fungal decay was associated with failure rates ranging from 75%–86% for *Q. agrifolia*, *Q. lobata*, and "other oaks" (*Q.* spp.) [15]. For *Q. agrifolia*, the most common type of failure was root and lower trunk rot in a residential setting. Across four cities in New York, USA, Luley et al. [16] found decay frequencies ranging from 53%–63% for three species of maple (*A. platanoides*, *A. saccharum*, and *A. saccharinum*). While, in Tampa, Florida, USA, Koeser et al. [17] determined decay frequency at 67% for *Q. laurifolia* and 29% for *Q. virginiana*. Across several northeastern states, Brazee and Marra [18] found a decay frequency of 30% for *Ulmus americana*. While decay frequencies can vary substantially among tree species, these studies illustrate that many landscape trees harbor internal decay.

Depending on the severity and location, internal decay can substantially decrease the strength of trees and increase the likelihood of failure if a tree experiences large loads. Practitioners have historically used a variety of tools to assess the severity and location of internal decay, from simple tools such as mallets for sounding a trunk to drills that measure wood resistance [19]. Previous work comparing the results of sonic and electrical resistance tomography to destructively sampled cross-sections has shown that tomography can accurately depict the internal condition of deciduous hardwoods [18,20]. Based on 105 stem disks cut from 72 trees where tomography scans were obtained, tomography correctly predicted the internal condition (no decay, incipient decay, active decay, and active decay with a cavity) in 95 cross-sections. For the remainder, the error occurred when small cavities (<10 cm in diameter) were misidentified [21].

In addition to accurately predicting a tree's internal condition, it has been shown that sonic tomograms can aid in understanding the loss in load-bearing capacity due to internal decay [22,23]. For decades, researchers have examined the use of various strength loss formulas, adapted from mechanics, to inform the management of decayed trees [24–26]. Using destructive samples, the percent reduction to the section modulus ($Z_{LOSS}$) was estimated numerically from sonic tomograms [22,23] and the estimates were found to be more accurate than other analytical strength loss formulas. However, the results have also shown tomography can underestimate the area of decay in smaller-diameter, circular trees and overestimate decay in larger-diameter, irregularly shaped trees [21–23]. Furthermore, the distance between the center of the decay and the center of the trunk, an important consideration in strength loss models, can be portrayed inaccurately. Despite these limitations, tomograms can provide additional insight when assessing strength loss due to decay, which is a critical component of tree risk assessments [19].

The primary goals of this study were to determine: (i) the incidence and severity of internal decay in the lower trunk of oak species commonly occurring in developed landscapes of the Northeast, USA; (ii) if the frequency and severity of decay are significantly different by numerous variables, including oak species, oak group (red vs. white), presence of decay symptoms, diameter, and sampling height, among others; and (iii) the similarity between strength loss values computed using the detailed information in tomograms (i.e., $Z_{LOSS}$) and others lacking such information (i.e., $I_{LOSS}$).

## 2. Materials and Methods

### 2.1. Tree Selection and Pathogen Identification

Oaks sampled for this study were non-randomly selected based on species, size, and access. Trees were sampled once for this study and data collection occurred over a six-year period (2016–2021) from approximately mid-April to early November to avoid subfreezing temperatures. This follows the manufacturer's recommendation, as subfreezing

temperatures can influence the accuracy of the tomograms. The sites spanned a variety of developed landscape settings, including urban and suburban streets, residential properties, public parks, and arboreta. No forest trees were sampled as part of this study. Roughly one-third of all oaks sampled (63/186; 34%) came from the Frank A. Waugh Arboretum on the University of Massachusetts, Amherst campus. With few exceptions, oaks were sampled from sites in Massachusetts, USA, where annual mean temperature and precipitation (1901–2000) are 8.3 °C and 113 cm, respectively [27]. Five oak species were sampled: *Q. alba* (white oak), *Q. bicolor* (swamp white oak), *Q. palustris* (pin oak), *Quercus rubra* (northern red oak), and *Q. velutina* (black oak).

Prior to tomography scanning, DBH was measured and symptoms and signs of internal decay were detected by visually assessing the lower trunk and the upper canopy of each tree. These areas were selected because tomography scanning was only conducted in the basal 2.5 m of the trunk. Symptoms of internal decay on the lower trunk included: excessive basal tapering (or flaring and swelling), depressions, cracks, seams, bulges, sap flow, dead bark, open cavities, and canopy dieback. Canopy dieback can be caused by various biotic (e.g., vascular wilt disease, insect pests, etc.) and abiotic (e.g., drought, mechanical root injury, etc.) stresses, but root damage from decay can manifest as symptoms in the upper portion of the canopy. Root decay is also frequently associated with decay in the root flare and lower trunk. Signs of a fungal decay pathogen included fruiting bodies or asexual structures. When signs were present, identification was made based on macroscopic features in the field and, when required, microscopic characters at the UMass Plant Diagnostic Laboratory.

### 2.2. Sonic and Electrical Resistance Tomography

To capture sonic and electrical resistance tomograms, the PiCUS® Sonic Tomograph 3 and TreeTronic 3 (Argus Electronic GMBH, Rostock, Germany) were used in this study. For specific details on the establishment of sampling cross-sections and the process of collecting sonic and electrical resistance tomograms, refer to Marra et al. [21]. For most trees, the lowest cross section was established close to the soil line, considering the geometry of the lower trunk, symptoms of decay (e.g., open cavities or decaying areas of outer sapwood), and location of perennial fruiting bodies of known wood-decaying fungi (if present). If the decay was detected based on the first scan, a second cross-sectional plane was established above the first sampling height. Often, but not always, the second plane was established 50 cm above the lower plane. If the decay was not detected during the first scan, no further sampling took place for most, but not all trees.

For each scan, galvanized roofing nails, 5.1–6.4 cm in length and spaced at 18–25 cm intervals, were then inserted to a depth just beneath the outer bark so the nail point contacts the sapwood. Each nail is a measuring point (MP) from which sonic and electrical resistance data are collected. The MPs are sequentially numbered with MP-1 placed at magnetic north. For all cross-sections, every attempt was made to use as many MPs as possible, to a maximum of 24, proportionally to the circumference of the cross-section. The height above ground, $H$ (cm), and diameter, $D$ (cm), of each measured cross section were recorded.

For sonic tomography (SoT), sensors were magnetically attached at each SoT MP and connected via cable to a detection module that is wirelessly connected to the PiCUS® software. At each MP, sound waves are initiated with sequential taps from the "sonic hammer" connected wirelessly to the detection module. The software then uses these data along with the inter-MP distances to calculate sonic velocities. The software then produces an image with a colorimetric scale depicting the internal condition within the cross-section. The colorimetric scale designates healthy, intact wood as brown (higher relative velocities) while cracked, damaged, decaying wood is designated by green, violet, and blue colors (lower relative velocities, in decreasing order).

For electrical resistance tomography (ERT), positive and negative leads are attached to each pair of previously established MP nails and connected via cable to a detection module connected wirelessly to the PiCUS® software. Upon user initiation from the software,

the detection module automates a process whereby, starting with one pair of leads and proceeding sequentially around the tree through each subsequent pair of leads, an electrical pulse is generated and detected by the other electrode pairs. Deviations from homogeneity in the wood result in a map of relative electrical conductivity, correlating principally with water content but also changes in ion concentration and/or cell structure. The ERT map uses red to portray areas of highest electrical resistance, progressing through orange, yellow, green, and blue with decreasing resistance.

### 2.3. Interpretation of Tomograms and Estimates of $A_D$ and $Z_{LOSS}$

Data from SoT and ERT were interpreted jointly using the PiCUS® Q74 Expert software to predict the internal condition at each sampled cross-section, based on the following criteria, slightly modified from Marra et al. [21]:

(A) Maximum sonic velocities and higher relative ER in the heartwood represent sound (non-decayed) wood. This condition appears as brown in the SoT and non-blue (yellow, orange, and red) in the ERT;

(B) Maximum sonic velocities and lower relative ER in the heartwood represent (i) incipient decay in which there is an increase in moisture content but reductions in wood density are not yet detectable; (ii) sound wood with an increased cation concentration in the heartwood and possibly a lower pH; and (iii) sound wood with an increased moisture content due to bacterial wetwood colonization. This condition appears brown in the SoT and blue in the ERT;

(C) Reduced sonic velocities and lower relative ER in the same corresponding location of the cross-section represent active fungal decay. This condition appears non-brown (green, violet, and blue) in the SoT and blue in the ERT;

(D) Reduced sonic velocities and the highest relative ER in the same corresponding location of the cross-section represent decay with a cavity. This condition appears violet and blue in the SoT and red in the ERT.

This interpretation strategy was based on earlier studies using destructive samples consisting of *Acer saccharum* (sugar maple), *A. rubrum* (red maple), *Betula alleghaniensis* (yellow birch), *B. lenta* (black birch), and *Fagus grandifolia* (American beech) [20,21], as well as guidelines provided by the manufacturer.

In the field, data checking following SoT allowed for the detection of abnormally high sonic velocities from individual MPs and subsequent correction. When evaluating the sonic tomograms, several changes to the default settings were considered. When the minimum velocity depicted was 50% of the maximum velocity (default setting), the maximum color space was expanded to determine the true minimum percent velocity value. When the minimum percent velocity is <40%–45%, moderate to significant changes in the area of decay ($A_D$) can develop and the color space remained expanded. Both SoT1 (default setting) and SoT2 calculation options were used to analyze each tomogram. However, the majority of sonic tomograms were generated using the SoT2 calculation. Line graphics were also examined at each MP along with estimates of internal cracks to select a final configuration. Finally, for trees with large areas of decay, and in some cases cavities, the "zero value correction" was used when the zero data warning was indicated by the software.

Based on the finalized tomograms, $A_D$ estimates (%) were recorded, and the reduction to the section modulus as a result of decay ($Z_{LOSS}$; %) was calculated. Two estimates of $A_D$ were recorded, one that included the green, violet, and blue ($A_D$-GVB) area of the sonic tomogram and one that included only violet and blue ($A_D$-VB). This distinction was made because the PiCUS® software excludes areas of green when differentiating solid and damaged wood. The exclusion of green can lead to significant differences in estimates of $A_D$ that may underestimate the true area of decay [21,22]. Based on destructive sampling, it has also been found that for smaller-diameter, circular-shaped trees, error can be reduced when estimates are based on $A_D$-GVB using the SoT1 calculation. However, for larger diameter, irregularly shaped trees, especially those with substantial decay present, estimates based on $A_D$-VB using the SoT2 calculation can reduce error [23]. Because trees

sampled here fall in between these two classifications (mostly circular-shaped but larger in diameter), careful interpretation was required. The numerical method *zloss* [28] was used to estimate the maximum percent $Z_{LOSS}$ for each tomogram with decay present and the offset length, $L_O$ (m), between the centroid of the trunk and the centroid of the largest damaged part in MATLAB (MathWorks, Natick, MA, USA); see Burcham et al. [22] for more detailed information about the estimation of $A_D$, $Z_{LOSS}$, and $L_O$ from tomograms. To express the offset length as a percent, $L_O$ was normalized by the radius of the measured section, approximated as $0.5D$.

### 2.4. Statistical Analyses

Chi-square goodness of fit, using expected values [29], was used to determine if there were significant differences in the frequency of decay symptoms and fungal decay pathogens by oak species and oak group. Groups were defined as "red oaks" (*Q. rubra*, *Q. velutina*, and *Q. palustris*; members of section *Lobatae*) and "white oaks" (*Q. alba* and *Q. bicolor*; members of section *Quercus*) [30]. Chi-square goodness of fit, using expected values, was also used to determine if there were significant differences in the frequency of internal decay by oak species, oak group, and the presence or absence of decay symptoms and fungal decay pathogens. One-way analysis of variance (ANOVA) was used to determine if mean DBH values were significantly different by oak species.

The relationship between 14 explanatory variables and the incidence and severity of decay was evaluated to select important variables for model development, which included: trunk diameter at 1.37 m above ground, *DBH* (cm); decay sampling height, *H* (cm); diameter at decay sampling height, *D* (cm); species identity (*Q. alba*, *Q. bicolor*, *Q. palustris*, *Q. rubra*, and *Q. velutina*); fungal pathogen (absent, present); decay symptoms (absent, present); taxonomic section (red oaks = *Lobatae*, white oaks = *Quercus*); basic green wood density, $\rho$ (g·cm$^{-2}$); shade, flood, and drought tolerance ratings [31]; and several mechanical properties [32], including modulus of elasticity, MOE (GPa), modulus of rupture, MOR (MPa), and work to maximum load, WML (kJ·m$^{-3}$).

The relative significance of each explanatory variable for predicting the incidence and severity of decay was evaluated using boosted regression trees (BRT). Based on the recommendations of Elith et al. [33], an optimal number of regression trees with three nodes, given the relatively small sample size, was determined by selectively decreasing the learning rate to produce greater than 1000 trees in the final model. Individual variables were considered influential and used for further model development if their relative influence exceeded the threshold proposed by Muller et al. [34], equal to total influence divided by the number of predictors ($100/14 = 7.14$). BRT models were fit using the gbm.step function in the R dismo package.

Subsequently, the incidence and severity of decay were modeled separately using a two-part conditional model. Many existing studies [35–37] have used a similar approach because the underlying observations are constrained to the unit interval between zero and one, often with a large proportion of zeros. In the first stage, the probability of decay occurring in each tree was modeled using binomial logistic regression, and, in the second stage, the severity of decay in affected trees was subsequently modeled using beta regression.

First, the probability of decay incidence was modeled as follows:

$$Y \sim binomial(n, p)$$
$$\eta = \beta_0 + \sum_{i=1}^{n} \beta_i x_i$$
$$E\left(\frac{Y}{1-Y}\right) = e^{\eta}$$

where the binomial distribution depicted the binary decay outcome *Y* for *n* observations with probability *p*, the linear predictor contained model coefficients $\beta_0 \ldots \beta_n$ for $x_1 \ldots x_n$ independent variables and the logit function linearized the relationship between the linear predictor and mean response. The assumed linear relationship between the logit and

continuous predictors was examined by testing an additional interaction term between each variable and its natural logarithm [38], and the possible existence of multicollinearity among predictors was examined using variance inflation factors (VIF). The extent of support in the data for candidate models containing unique combinations of influential explanatory variables was examined using the AIC, and the model with the lowest AIC was selected for its relative parsimony and goodness of fit. For the final, reduced model, the goodness of fit was assessed using the model chi-square and area under the receiver operating characteristic (ROC) curve [38]; and the fit was diagnosed by inspecting plots of the change in Pearson chi-square, change in deviance, and Cook's distance against the estimated probability for each variable combination [38].

Second, the severity of decay was modeled, using $A_D$ determined using violet and blue (VB) and green, violet, and blue (GVB) separately, as follows:

$$Y \sim beta(\mu, \varphi)$$
$$\eta = \beta_0 + \sum_{i=1}^{n} \beta_i x_1$$
$$E\left(\frac{Y}{1-Y}\right) = e^{\eta}$$

where the Beta distribution depicted the extent of decay with mean $\mu$ and precision $\varphi$ [39], and the linear predictor and link function were the same as the logistic model. Alternative link functions commonly associated with the binomial distribution family (e.g., Poisson, log-log) were evaluated using log-likelihood statistics, but the logistic link best fit the data. The existence of heteroscedasticity in $A_D$ was detected by inspecting the significance, as measured by partial Wald tests, of variables used to separately model the precision parameter $\varphi$, and multicollinearity among predictors was evaluated using VIFs. After considering all combinations of influential explanatory variables, the candidate model with the lowest AIC was selected as the final, reduced model. Model fit was evaluated using mean bias (predicted − observed outcome) and root mean square error (RMSE), and the fit was diagnosed by inspecting plots of standardized weighted residuals and generalized leverage against predicted values [40]. The binomial logistic regression and beta regression models were fit using the glm and betareg functions, respectively, in R (R Core Team, Vienna, Austria, 2022).

To explore the mechanical implications of the modeled decay, $Z_{LOSS}$ estimated from tomograms was compared with strength loss estimates computed using $A_D$ obtained from the hurdle model. Since the empirically modeled $A_D$ depicted the size of decayed areas and not the shape, strength loss was computed using a basic formula approximating the shape of decayed trees as a hollow pipe [41]:

$$I_{LOSS} = \left(\frac{d}{D}\right)^4 \times 100$$

where $d$ and $D$ are the diameters of the circular equivalent of the decayed area and trunk at measurement height, respectively. The two strength loss estimates use related section properties (second moment of area, $I$, and section modulus, $Z$) to quantify a decayed tree part's reduced load-bearing capacity, but some have reasoned that $Z$ is a more appropriate measure because it directly relates an applied bending moment to the maximum bending stress experienced by a beam. For $I_{LOSS}$, the related equation is solved symbolically by framing the problem in a simplified form—the shape of the decayed section is approximated as a hollow pipe. For $Z_{LOSS}$, the related equation is solved numerically without any simplifying assumptions about the decayed tree part's geometry. Although $I_{LOSS}$ can be computed easily in a few steps, the additional steps needed to accommodate greater geometric detail for $Z_{LOSS}$ can only be practically accomplished using a computer. Using $Z_{LOSS}$ and $I_{LOSS}$ determined from $A_D$-VB and $A_D$-GVB separately, the distance between the two strength loss estimates was computed as the actual difference between the two percentages. It was expected that the distance between strength loss estimates would increase for biased empirical estimates of $A_D$ with the predicted decay severity dissimilar

from the corresponding tomographic measurements of the same tree and offset decayed areas poorly approximated as concentric circles in the geometrically simplified $I_{LOSS}$ equation. The hypothesized relationship was tested using a linear regression model containing residuals from the beta regression models and the product of $A_D$ and $L_O$, a combined measure of offset decay. Plots of studentized residuals against fitted values were inspected for homoskedasticity and linearity, and Cook's D was used to identify potential outliers, with cases exerting influence greater than $4/n$ inspected more closely. The linear model was fit using the lm function in R (R Core Team, 2022).

## 3. Results

### 3.1. Tree Characteristics

Mean DBH values ranged from 93–104 cm across the five oak species and were not significantly different from one another (F = 1.12, $p$ = 0.351; Table 1). Symptoms of internal decay were present on 75 of 186 (40%) oaks sampled (Table 1). The frequency of decay symptoms was significantly higher for *Q. rubra* compared to expected values, but there were no other significant differences by species or oak group (Table 1). Fruiting bodies and/or asexual structures of a fungal decay pathogen were present on 50 of 186 (27%) sampled oaks (Table 1). The frequency of a decay pathogen was significantly higher for *Q. rubra* and lower for *Q. alba* compared to expected values (Table 1). As a result, among oak groups, the frequency of a decay pathogen was significantly higher for the red oak group and lower for the white oak group (Table 1). Ten species of wood-decaying fungal pathogens from seven genera were encountered in this study and are listed in Table 2.

**Table 1.** The number of trees sampled, mean and range of DBH values, and frequency of decay symptoms and fungal decay pathogens for each oak species and oak group.

| | | DBH (cm) | | Decay Symptoms | | | | Fungal Decay Pathogen | | | |
|---|---|---|---|---|---|---|---|---|---|---|---|
| **Oak Species** | **n** | **Mean** | **Range** | **Present** | **Absent** | $\chi^2$ | ***p*-Value** | **Present** | **Absent** | $\chi^2$ | ***p*-Value** |
| *Q. rubra* | 44 | 104 | 60–194 | **25** ↑ | **19** ↓ | **4.607** | **0.032** | **22** ↑ | **22** ↓ | **11.458** | **<0.001** |
| *Q. palustris* | 35 | 101 | 49–182 | 9 | 26 | 2.976 | 0.084 | 9 | 26 | 0 | 1.000 |
| *Q. velutina* | 41 | 93 | 51–202 | 19 | 22 | 0.402 | 0.526 | 15 | 26 | 1.988 | 0.159 |
| *Q. alba* | 43 | 100 | 52–154 | 16 | 27 | 0.097 | 0.755 | **4** ↓ | **39** ↑ | **7.398** | **0.007** |
| *Q. bicolor* | 23 | 95 | 52–154 | 6 | 17 | 1.643 | 0.200 | 0 | 23 | n/a | n/a |
| **Oak Group** | | | | | | | | | | | |
| Red Oak | 120 | 100 | 49–202 | 53 | 67 | 0.868 | 0.351 | **46** ↑ | **74** ↓ | **8.352** | **0.004** |
| White Oak | 66 | 98 | 52–154 | 22 | 44 | 1.567 | 0.211 | **4** ↓ | **62** ↑ | **14.972** | **<0.001** |
| Total | 186 | 99 | 49–202 | 75 | 111 | | | 50 | 136 | | |

Values in bold indicate significant differences based on Chi-square analysis (using expected values) at $p$ = 0.05. Arrows denote if the value is significantly higher (↑) or lower (↓) than the expected value.

**Table 2.** Fungal wood-decaying pathogens identified from sampled oaks.

| **Wood Decay Pathogen** | **n** |
|---|---|
| *Grifola frondosa* | 14 |
| *Ganoderma* spp. | 11 |
|     *G. applanatum* (1) | – |
|     *G. curtisii* (1) | – |
|     *G. sessile* (9) | – |
| *Laetiporus* spp. | 10 |
|     *L. cincinnatus* (2) | – |
|     *L. sulphureus* (5) | – |
|     *L.* sp. (3) | – |
| *Armillaria* sp. | 8 |
| *Niveoporofomes spraguei* | 8 |

**Table 2.** *Cont.*

| Wood Decay Pathogen | n |
|---|---|
| *Bondarzewia berkeleyi* | 4 |
| *Climacodon septentrionalis* | 1 |
| **Total** | 56 |

The number of individual species identified appears in parentheses.

### 3.2. Internal Condition and $A_D$

Overall, 323 pairs of sonic and ER tomograms were obtained from 186 oaks, with the number of sampled trees by oak species ranging from 23–44 (Table 3). Of the 186 oaks sampled, 135 (73%) had detectable decay within the lower trunk, while 51 (27%) did not (Table 3). Of the 135 oaks with decay, 72 of 135 (53%) exhibited low ER (higher relative conductivity) in the same area of the cross-section where decay was found, indicating the decaying wood tissue was still present (Figures 1 and S1). Meanwhile, 63 of 135 (47%) oaks had areas of high ER (lower relative conductivity) in the same area where decay was detected, indicating that cavity formation was likely occurring (Figures 2 and S2–S4). Chi-square analysis determined that there were no significant differences in the frequency of decay incidence by oak species (Table 3). However, the white oak group exhibited a lower frequency of internal decay incidence compared to expected values (Table 3). The frequency of internal decay was significantly higher for trees with visible symptoms while it was significantly lower when symptoms were absent (Table 3). Finally, internal decay frequency was significantly higher when fruiting bodies were present but no significant differences were found in decay frequency when fruiting bodies were absent (Table 3). When all sonic tomograms are evaluated together, 229 of 323 (71%) had measurable decay present while in 94 of 323 (29%) decay was absent. Across all oak species, the mean $A_D$ was 41% ($A_D$-GVB) and 31% ($A_D$-VB), respectively, while the mean maximum $Z_{LOSS}$ was 35% ($Z_{LOSS}$-GVB) and 22% ($Z_{LOSS}$-VB), respectively (Table S1). By oak species, mean $A_D$ ranged from 35%–47% ($A_D$-GVB) and 22%–37% ($A_D$-VB), while mean $Z_{LOSS}$ ranged from 26%–40% ($Z_{LOSS}$-GVB) and 18%–26% ($Z_{LOSS}$-VB) (Table S1). The mean $A_D$ and mean maximum $Z_{LOSS}$ by scanning height for each oak species can be found in Table S1.

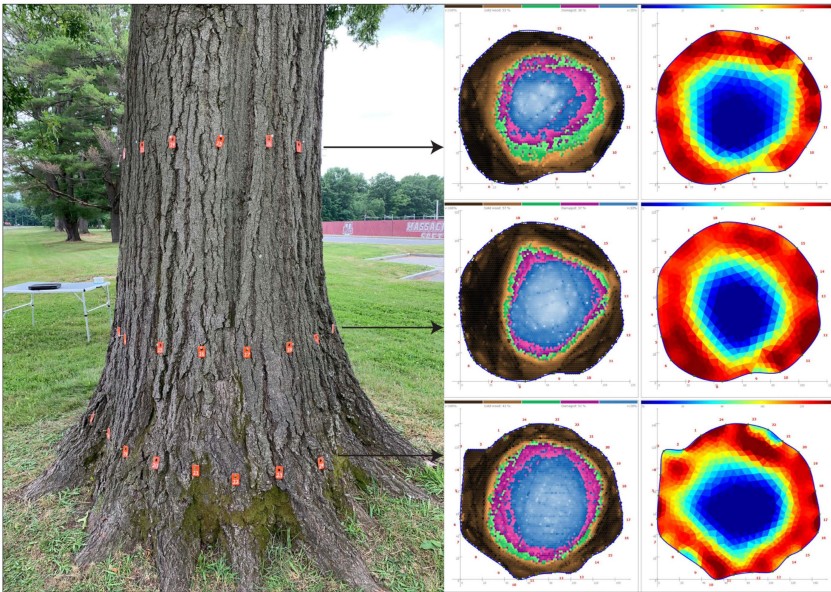

**Figure 1.** A pin oak (*Quercus palustris*) in the Waugh Arboretum at the University of Massachusetts, Amherst. The sonic tomograms (**middle**) depict decay in the heartwood while the electrical resistance tomograms (**right**) show low electrical resistance (high conductivity). Interpreted together, they indicate the decaying wood is still present and a cavity has not yet formed.

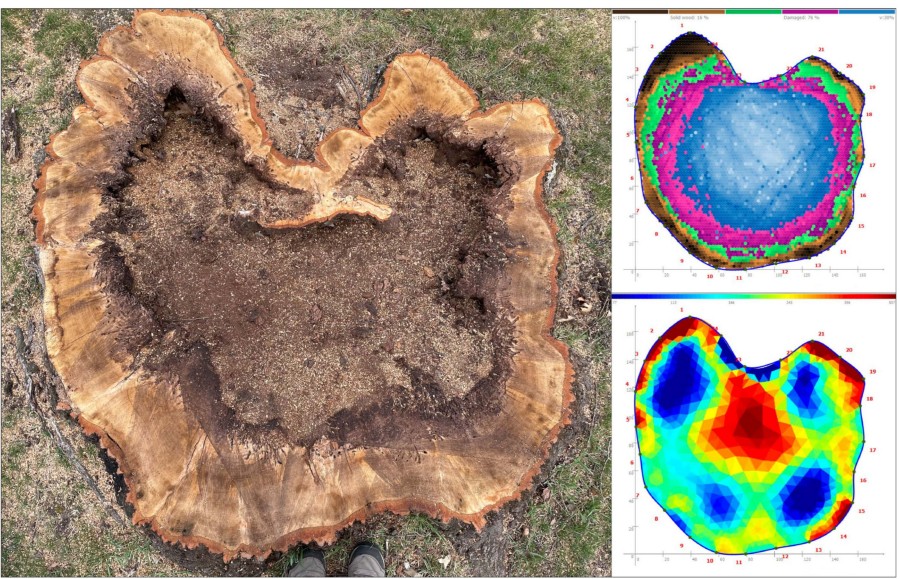

**Figure 2.** Overhead view showing the stump of a northern red oak (*Quercus rubra*) that suffered butt rot from *Armillaria*. The sonic tomogram (**upper right**), captured at a height of 40 cm from the soil line, depicted extensive decay within the heartwood. The electrical resistance tomogram (**bottom right**) depicted high electrical resistance (low conductivity) within the decaying heartwood. When interpreted together, they indicate a cavity had developed, which was corroborated when the tree was removed.

**Table 3.** Frequency of internal decay by oak species, oak group, and presence of decay symptoms and fungal decay pathogen.

| | | Internal Decay | | | |
|---|---|---|---|---|---|
| **Oak Species** | **n** | **Present** | **Absent** | $\chi^2$ | *p*-**Value** |
| *Q. rubra* | 44 | 37 | 7 | 2.865 | 0.091 |
| *Q. palustris* | 35 | 25 | 10 | 0 | 1.000 |
| *Q. velutina* | 41 | 34 | 7 | 1.988 | 0.159 |
| *Q. alba* | 43 | 26 | 17 | 2.890 | 0.089 |
| *Q. bicolor* | 23 | 13 | 10 | 3.608 | 0.058 |
| Oak Group | | | | | |
| Red Oak | 120 | 96 | 24 | 3.386 | 0.066 |
| White Oak | 66 | **39 ↓** | **27 ↑** | **6.188** | **0.013** |
| Decay Symptoms | | | | | |
| Present | 75 | **69 ↑** | **6 ↓** | **14.881** | **<0.001** |
| Absent | 111 | **66 ↓** | **45 ↑** | **10.278** | **0.001** |
| Fungal Decay Pathogen | | | | | |
| Present | 50 | **44 ↑** | **6 ↓** | **6.349** | **0.012** |
| Absent | 136 | 91 | 45 | 2.376 | 0.123 |
| Total | 186 | 135 | 51 | | |

Values in bold indicate significant differences based on Chi-square analysis (using expected values) at *p* = 0.05. Arrows denote if the value is significantly higher (↑) or lower (↓) than the expected value.

Using the threshold criterion, boosted regression trees indicated that five variables were highly influential for predicting the incidence of decay in measured sections, including (ranked in terms of decreasing relative influence) *D*, *DBH*, *H*, symptoms, and species (Table 4). The same set of variables, except for symptoms, was highly influential for predicting the severity of decay determined using either tomogram color set. The ranking

of variables differed slightly for predicting $A_D$-GVB and $A_D$-VB, but $D$, $DBH$, and $H$ were consistently more influential than species.

**Table 4.** The relative influence of variables used to predict the incidence and severity of decay.

| | Relative Importance | | |
|---|---|---|---|
| **Variable** | **Incidence** | $A_D$**-GVB** | $A_D$**-VB** |
| Basic density, $\rho$ (g·cm$^{-3}$) | 2.3 | 0.7 | 0.7 |
| Diameter at breast height, $DBH$ (cm) | 17.1 * | 20.0 * | 33.5 * |
| Diameter at measurement height, $D$ (cm) | 34.9 * | 31.1 * | 26.8 * |
| Drought Tolerance, DT | 0.4 | 0.9 | 0.6 |
| Flood Tolerance, FT | 1.9 | 1.6 | 1.3 |
| Measurement height, $H$ (cm) | 12.8 * | 20.2 * | 14.0 * |
| Modulus of Elasticity, MOE (GPa) | 1.1 | 1.6 | 1.5 |
| Modulus of Rupture, MOR (kPa) | 0.1 | 0.0 | 0.1 |
| Pathogen | 4.2 | 5.0 | 3.8 |
| Shade Tolerance, ST | 0.6 | 0.6 | 0.7 |
| Species | 10.4 * | 10.6 * | 10.9 * |
| Symptoms | 12.5 * | 4.4 | 3.8 |
| Taxonomic Section | 0.4 | 0.2 | 0.1 |
| Work to Maximum Load, WML (kJ) | 1.3 | 2.9 | 2.2 |

* denotes variables selected as influential with relative influence greater than expected by chance (7.14).

Using the reduced set of highly influential variables, logistic and beta regression models were fit to predict decay incidence and severity, respectively. However, $DBH$ and $D$ were highly correlated ($r = 0.87$), and the corresponding terms in a model containing both variables had a VIF exceeding 9. In most cases, $DBH$ was less influential than $D$ for predicting decay incidence and severity, and it was removed from the list of candidate variables for model selection since $DBH$ can be considered a special case of $D$. Using information criteria, the reduced binomial logistic regression model containing $D$, symptoms, and species were selected for predicting the incidence of decay (Table 5; Figure 3). Diagnostic plots suggested that the model poorly fit eight observations associated with five variable combinations, but the change in model coefficients after excluding the observations was similar to other cases, indicating their limited influence. Generally, the observations defied model expectations by containing a combination of factors that should have produced decay but did not, especially for two large trees exhibiting symptoms of decay. Despite the poor case-wise fit, the observations were retained in the model to accurately depict the sample of large, mature oaks.

**Table 5.** Model coefficients and corresponding effect sizes for the binomial logistic regression model fit to binary decay incidence in landscape oaks.

| Term | Coefficient (SE) | $p$ | Odds Ratio (95% CI) | Average Marginal Effects or Means (95% CI) |
|---|---|---|---|---|
| Intercept | −2.24 (0.73) | 0.002 | 0.11 (0.02–0.43) | |
| $D$ | 0.26 (0.06) | <0.001 | 1.30 (1.17–1.47) | 0.05 (0.03–0.06) |
| Symptoms (present) | 1.19 (0.29) | <0.001 | 3.30 (1.88–5.93) | 0.21 (0.12–0.30) |
| Species (*Q. rubra*) | - | - | - | 0.81 (0.73–0.90) [ab] |
| Species (*Q. palustris*) | −0.78 (0.41) | 0.056 | 0.46 (0.20–1.01) | 0.68 (0.56–0.79) [a] |
| Species (*Q. velutina*) | 0.37 (0.43) | 0.393 | 1.44 (0.62–3.39) | 0.86 (0.79–0.94) [b] |
| Species (*Q. alba*) | −0.58 (0.41) | 0.161 | 0.56 (0.25–1.25) | 0.72 (0.61–0.82) [ab] |
| Species (*Q. bicolor*) | −0.79 (0.46) | 0.088 | 0.45 (0.18–1.12) | 0.67 (0.53–0.82) [ab] |

Model coefficients, presented in the link (logit) scale, depict change in log-odds over a 10 cm increase in $D$ and, for categorical variables, compared to the reference level (symptoms absent, *Q. rubra*). The average marginal effects or means are presented in the response (probability) scale. Computed by averaging over both levels of symptoms (absent, present) at the average $D$ (117 cm), the average marginal means of each species followed by the same letter are not significantly different at the $\alpha = 0.05$ level. Model $\chi^2(3) = 54.33$, $p < 0.01$; AUC = 0.75; n = 323.

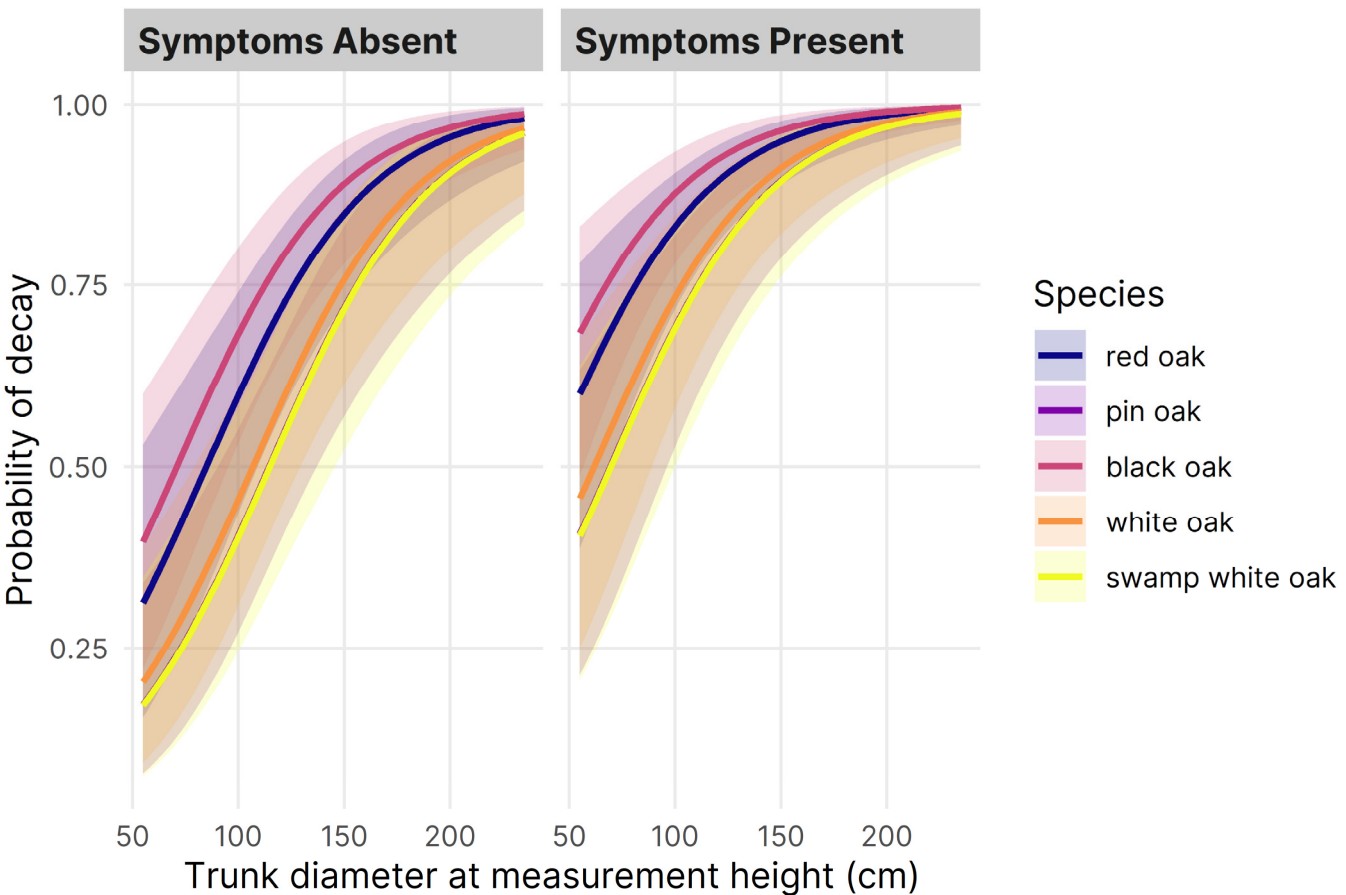

**Figure 3.** The probability of decay for oak species (*Q. rubra*, *Q. palustris*, *Q. velutina*, *Q. alba*, and *Q. bicolor*) by diameter at sampling height (cm) and presence of symptoms predicted by a binomial logistic regression model fit to binary decay incidence.

The area under the ROC curve (AUC = 0.75) indicated acceptable model discrimination between solid and decayed trees, and the model coefficients generally depicted a higher probability of decay for large trees with symptoms of decay: the odds of decay occurrence were 1.3 and 3.3 times greater over a 10 cm increase in *D* and for symptomatic compared to asymptomatic trees, respectively (Table 5). Relative to *Q. rubra* (the reference species), the model coefficients depicted a lower probability of decay for all other species, except *Q. velutina*. Apart from the significant difference in the probability of decay for *Q. palustris* (lower) and *Q. velutina* (higher), the marginal means of the probability of decay for each species were not significantly different from one another (Table 5). Although it had one of the lowest mean values, *Q. bicolor* showed marked variability in the incidence of decay compared to the other modeled species. The confidence intervals surrounding predicted values depicted moderate uncertainty for most variable combinations, but the smaller intervals for large, symptomatic trees depicted the increasing certainty of decay in such cases (Figure 3).

Using information criteria, the reduced beta regression model containing *D*, *H*, and species was selected for predicting the severity of the decay, regardless of the colors (GVB, VB) used to determine $A_D$. There were no outliers or influential observations detected for the model fit to $A_D$-GVB, but several $A_D$-VB observations exerted undue influence on model coefficients. In six cases, the decayed areas in tomograms were mostly displayed using green and limited violet or blue, and the related observations with $A_D$-VB effectively equal to zero were removed to improve model coefficients. After removing the observations, the fit statistics indicated a satisfactory description of the data, with a mean bias below 1% for

models fit to $A_D$-GVB and $A_D$-VB, but the prediction bias varied considerably between −50% and 40% for individual cases in both models (Table 6).

**Table 6.** Model coefficients and corresponding effect sizes for beta regression models fit to decay severity obtained from sonic tomography of landscape oaks.

| | **Model Coefficients** | | | | **Average Marginal Effects or Means** | |
| | $A_D$-GVB | | $A_D$-VB | | $A_D$-GVB | $A_D$-VB |
| **Term** | **Estimate (SE)** | *p* | **Estimate (SE)** | *p* | **Estimate (95% CI)** | **Estimate (95% CI)** |
|---|---|---|---|---|---|---|
| Mean, $\mu$ | | | | | | |
| Intercept | −0.88 (0.27) | 0.001 | −1.60 (0.30) | <0.001 | | |
| *D* | 0.07 (0.02) | <0.001 | 0.09 (0.02) | <0.001 | 1.6 (0.8, 2.5) | 1.8 (1.0, 2.5) |
| *H* | −0.06 (0.02) | <0.001 | −0.06 (0.02) | <0.001 | −1.4 (−2.1, −0.7) | −1.2 (−1.8, −0.5) |
| Species (*Q. rubra*) | - | | - | | 40.7 (36.1, 45.4) [ab] | 29.7 (25.4, 34.0) [ab] |
| Species (*Q. palustris*) | −0.35 (0.17) | 0.032 | −0.22 (0.18) | 0.221 | 32.5 (26.8, 38.2) [a] | 25.3 (20.0, 30.7) [ab] |
| Species (*Q. velutina*) | 0.05 (0.14) | 0.749 | 0.03 (0.15) | 0.869 | 41.8 (36.9, 46.8) [ab] | 29.1 (24.6, 33.7) [ab] |
| Species (*Q. alba*) | 0.13 (0.15) | 0.392 | 0.17 (0.16) | 0.292 | 43.9 (38.3, 49.6) [b] | 33.3 (28.0, 38.6) [b] |
| Species (*Q. bicolor*) | −0.30 (0.20) | 0.137 | −0.52 (0.23) | 0.026 | 33.7 (25.9, 41.5) [ab] | 20.1 (13.6, 26.7) [a] |
| Precision, $\varphi$ | | | | | | |
| Intercept | 2.56 (0.34) | <0.001 | 2.44 (0.35) | <0.001 | | |
| *D* | −0.07 (0.03) | 0.009 | −0.6 (0.03) | 0.026 | | |

Diameter at scanning height, $D$ (m $\times 10^{-1}$), scanning height, $H$ (m$\times 10^{-1}$), and species (*Q. rubra, Q. palustris, Q. velutina, Q. alba, Q. bicolor*), were used to estimate the mean decay severity determined using either green, violet, and blue ($A_D$-GVB, $n = 229$) or violet and blue ($A_D$-VB, $n = 223$) in tomograms; $D$ was also used to model differences in the precision (inverse of variability) of decay severity observed at different cross-sectional sizes. Fit statistics were similar for the models fit to $A_D$-GVB (mean bias = −0.48%; RMSE = 19.36%) and $A_D$-VB (mean bias = 0.14%; RMSE = 19.46%). The model coefficients, presented in the link (mean: logit; precision: log) scale, depict the change in log(it)-transformed $A_D$ over a 10 cm increase in $D$ or $H$ and, for categorical variables, compared to the reference level (*Q. rubra*). The average marginal effects or means are presented in response (percent) scale. Computed using the average value of $D$ (mean: 122 cm) and $H$ (mean: 60 cm), the average marginal means of each species followed by the same letter are not significantly different at the $\alpha = 0.05$ level.

Regardless of the colors used to determine $A_D$, the model coefficients generally indicated that decay severity increased for large sections near the ground, and the species terms indicated that the severity of decay was intermediate for *Q. rubra* compared to the other species (Table 6). The average marginal effects of $D$ and $H$, indicating a 1%–2% change in the severity of decay over a 10 cm increase in the diameter or height of the scanned section, were similar in magnitude but oppositely signed. Due to the different color sets used to measure decay severity, the average marginal means for each of the species varied between the $A_D$-GVB and $A_D$-VB datasets, and the ranking of average marginal mean $A_D$ among species was reordered slightly between the two datasets, except for the consistently largest mean $A_D$ in *Q. alba*.

Among all tomograms, the maximum $Z_{LOSS}$ determined using both color sets (GVB, VB) varied between 0% and 92% (mean: 29%), and $L_O$ for the decayed areas similarly depicted in all tomograms varied between 0% and 26% (mean: 4%). Among all observations, the distance between the two strength loss estimates ($Z_{LOSS} - I_{LOSS}$) varied between −24% and 76% (mean: 16%). The linear model fit to the distance between $I_{LOSS}$ and $Z_{LOSS}$ containing $A_D$ prediction errors and $A_D \cdot L_O$ was significantly better than a null model (F = 163.5; df = 2, 449; *p* < 0.001) containing only an intercept, affirming the hypothesized relationship. However, residual plots showed evidence of heteroscedasticity, confirmed using the studentized Breusch–Pagan test (*p* < 0.001), and a quadratic relationship between the independent and dependent variables. After inspection, the model was re-fit with a quadratic term for the residuals variable and a non-constant variance estimator to obtain robust standard errors (Table 7).

**Table 7.** Model coefficients and corresponding effect sizes for linear regression model fit to the distance between two strength loss estimates.

| | Model Coefficients | | Average Marginal Effects | |
|---|---|---|---|---|
| **Term** | **Estimate (SE)** | ***p*** | **Values** | **Estimate (95% CI)** |
| Intercept | −4.0 (1.0) | <0.001 | | |
| $(A_D \times L_O)$ | 1456.8 (64.1) | <0.001 | | 14.57 (12.94, 16.19) |
| $(\text{Residuals})^2$ | 131.1 (31.1) | <0.001 | Residuals: | |
| | | | −40 | −10.49 (−13.59, −7.38) |
| | | | 0.6 | 0.16 (0.11, 0.21) |
| | | | 41.8 | 10.96 (7.72, 14.21) |

Note: The average marginal effects of the residuals term were computed at the observed minimum, mean, and maximum values, since it was quadratically related to the response. $R^2_{ADJ} = 0.419$.

The model coefficients showed that the distance between strength loss estimates increased proportional to the offset decayed area in tomograms and varied quadratically with the difference between the modeled and measured $A_D$ (Table 7; Figure 4). The average marginal effect of offset decay indicated that strength loss estimates were increasingly dissimilar as $A_D \cdot L_O$ increased, but the average marginal effect of residuals, calculated over the range of observed values, illustrated the increasing dissimilarity between strength loss estimates for progressively large positive and negative residuals. The distance between strength loss estimates was minimized when decayed areas occupied the center of the stem and empirical models accurately predicted $A_D$ in tomograms (Figure 4).

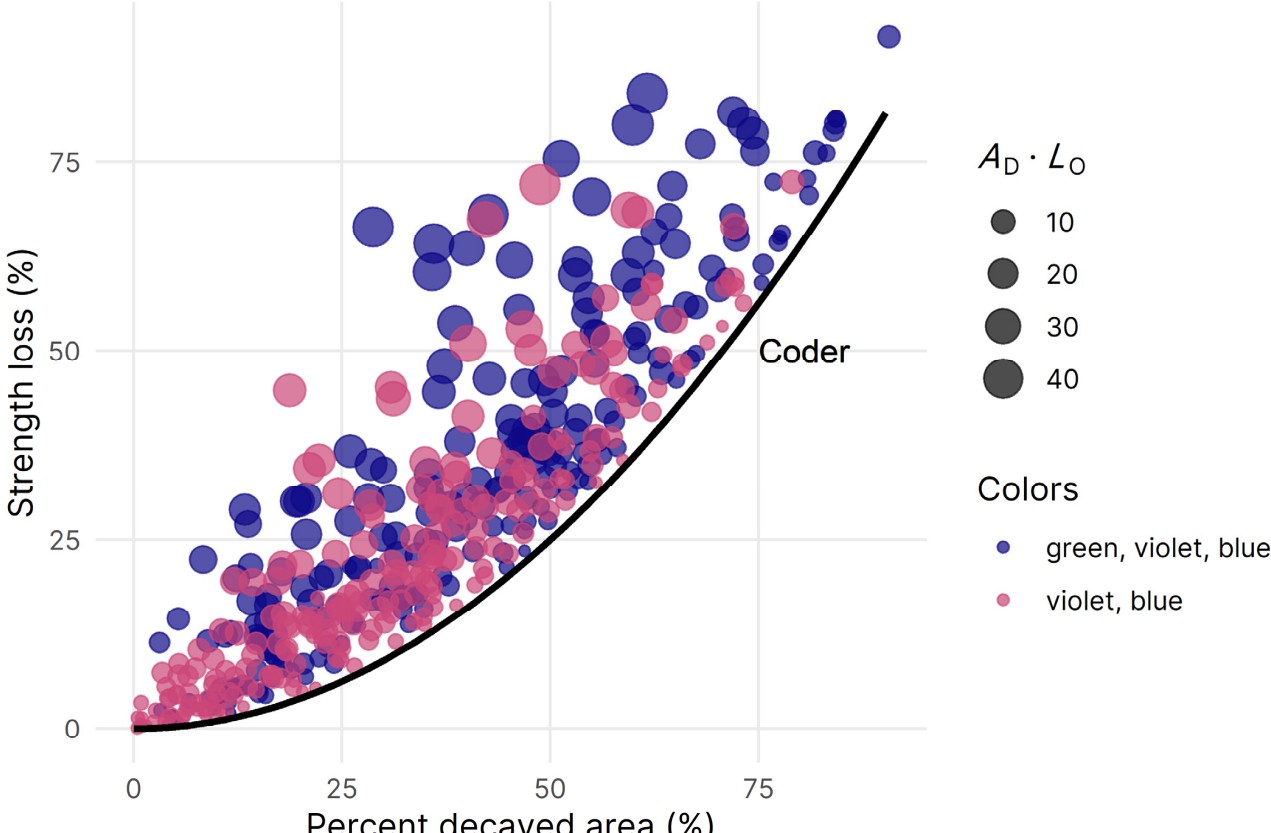

**Figure 4.** Scatterplot showing the relationship between strength loss estimates and percent area decayed for each tomogram with decay (n = 229). $Z_{LOSS}$ (Circles) and Coder's $I_{LOSS}$ (black line) estimates were made using $A_D$-GVB (green, violet, and blue) and $A_D$-VB (violet and blue), respectively. Offset areas of decay are depicted using a combined measure ($A_D \cdot L_O$) with $L_O$ representing the offset distance of the decay from the center of the trunk.

### 3.3. Sound Wood with Low Electrical Resistance

Among all sonic tomograms, 94/323 (29%) exhibited maximum sonic velocities throughout the sampled cross sections, indicating sound wood and no decay present. Within this dataset, low electrical resistance throughout the heartwood was present in 83/94 (88%) of all cases. In 8/11 remaining ER tomograms, a mixture of high and low resistance was present throughout the heartwood while in three cases, high ER dominated.

## 4. Discussion

### 4.1. Decay Incidence and Severity

Overall, the incidence of internal decay in the lower trunk was high among sampled oaks, with >70% having detectable decay present. The results showed important differences in the incidence of decay among the five oak species, but the considerable uncertainty around estimates reflected the importance of other determinants for decay incidence, apart from species. These findings are consistent with prior studies that have documented high levels of variability in decay incidence and severity across sites, ages, and tree species [11,21,42]. However, these results are also aligned with the longstanding recognition of differences in decay resistance among tree species, including oaks [10,43].

The boosted regression trees demonstrated the importance of diameter, height, symptoms, and oak species in predicting the incidence of decay, while the reduced binomial logistic regression model showed that only diameter, symptoms, and species were significant. Overall, *Q. rubra* and *Q. velutina* had the highest incidence, with >80% of sampled trees in each species having detectable decay. Yet, the probability of decay in *Q. rubra* was not significantly different from any other oak species. Further, with *Q. rubra* as the reference species, only *Q. velutina* and *Q. palustris* demonstrated significant differences in decay incidence. Ultimately, these results indicate that arborists can be reasonably confident about the likely existence of decay on large, symptomatic trees, regardless of the species. As shown in other studies [16–18], diameter proved to be one of the most important variables in predicting both the incidence and severity of internal decay.

There was substantial variation in the severity of decay among sampled oaks with $A_D$ values ranging from 2%–88%, high standard deviations in mean $A_D$ values, and a large range of confidence intervals in the average marginal effects. Yet, despite the variation, there was a significant decrease in mean $A_D$ with increasing height from the soil line across all oak species. The reduced beta regression model showed that height was a significant variable in predicting decay severity, regardless of whether intermediate velocities were used to determine $A_D$. These findings have important implications for studies assessing the incidence and severity of internal decay in landscape trees. Many of the fungal decay pathogens attacking oak, such as *Armillaria*, *Ganoderma*, *Grifola*, and *Laetiporus* (see Table 2) often establish in the roots and progress to the lower trunk of the tree [3,8].

Among the 135 oaks with decay, nearly half exhibited no visible symptoms of internal decay upon visual assessment. In this study, we were unable to determine the identity of every fungal pathogen associated with a decaying tree, which if possible, could have helped to explain the variation in severity [3]. While the results here show that height was ultimately not significant in determining decay incidence, if sampling is performed too high on the trunk, decay severity can be underestimated. These results also demonstrate the need to sample trees that show no symptoms of internal decay. When decay symptoms were absent, both Luley et al. [16] and Koeser et al. [17] performed resistance drilling to detect internal decay at 150 cm above the soil line. Because decay severity is often highest close to the soil line, sampling large diameter, asymptomatic trees at heights >120 cm could reveal only minor amounts of internal decay.

When all sonic tomograms with decay are evaluated, the inclusion of intermediate sonic velocities (green) increases mean $A_D$ by 10% compared to estimates that exclude intermediate velocities ($A_D$-GVB = 41% vs. $A_D$-VB = 31%). Furthermore, when sonic tomograms captured only at the lowest sample heights are considered, the location where decay incidence and severity were higher, the difference in mean $A_D$ is the same ($A_D$-GVB = 45%

vs. $A_D$-VB = 35%). This 10% difference is modest but could have major implications for how sonic tomograms are interpreted by arborists and their clients. Because the trees were not destructively sampled, the results of SoT and ERT cannot be corroborated against stem disks, as previous studies have allowed [21–23]. However, given that tree preservation is often a primary goal when carrying out tomography scans, arborists will often be unable to corroborate the results to actual stem disks in the field. However, by presenting both "liberal" ($A_D$-GVB) and "conservative" ($A_D$-VB) values, arborists can report the range of possible conditions in the lower trunk and incorporate these results into a thorough risk assessment report [19].

### 4.2. Red Oaks vs. White Oaks

Anecdotally, many foresters and arborists consider the white oak group to be more resistant to decay compared to the red oak group. *Quercus alba* produces abundant tyloses and tannins that are known to resist fungal pathogens, helping to make the wood prized for manufactured products [44,45]. In controlled trials using inoculated heartwood blocks cut from seven oak species (which included *Q. alba*, *Q. bicolor*, *Q. rubra*, and *Q. velutina*) it was found that members of the white oak group were significantly more decay-resistant compared to members of the red oak group [43]. However, according to Hepting [10], the reputation *Q. alba* has for superior decay resistance is not absolute, as several forest studies have shown high rates of decay for this species. Further, Highley [46] determined that untreated sapwood and heartwood of *Q. rubra* were equally as decay-resistant compared to *Q. alba*, with both rated as "most resistant" in a comparison of 19 species of conifers and hardwoods.

In this study, the white oak group exhibited a significantly lower frequency of internal decay incidence and the presence of wood decay pathogens compared to the red oak group. This difference was especially pronounced when directly comparing *Q. alba* to *Q. rubra*. Yet, the binomial regression showed when *Q. rubra* is used as the reference species its incidence of decay was not significantly different from any other species. Further, *Q. alba* exhibited the highest mean $A_D$ and mean maximum $Z_{LOSS}$ of any oak species. Yet, a comparison of mean $A_D$ values among oak species showed no clear differences among the white oak and red oak groups. So while the overall incidence was lower for the white oak group, decay severity for *Q. alba* was comparable to the members of the red oak group studied here. Comparatively, *Q. rubra* had the highest overall frequency of internal decay and was the only oak species with a significantly higher frequency of both decay symptoms and fungal decay pathogens present.

Surprisingly, *Q. palustris* had a significantly lower incidence of decay compared to *Q. velutina* and the lowest mean $A_D$ and mean maximum $Z_{LOSS}$ across all oaks sampled. While mean $A_D$ and maximum $Z_{LOSS}$ values were only statistically different from *Q. alba* when intermediate sonic velocities are included, they at least pose the question of whether *Q. palustris* may have greater decay resistance compared to oak species frequently planted in developed landscapes in Massachusetts. Based on phylogenetic classification of North American oaks, *Q. palustris* is a member of subsection *Palustres*, making this species genetically divergent from *Q. rubra* and *Q. velutina*, which are both members of subsection *Coccineae* [30]. Nevertheless, differences in decay severity could be explained by other factors, as oaks have previously demonstrated varying levels of decay resistance by region in the eastern U.S. [10].

### 4.3. Strength Loss Estimates

By accounting for the irregular shape of the decayed area and sampled cross-section, along with the offset occurrence of decay in some trees, strength loss estimates made using $Z_{LOSS}$ were on average 16% higher than estimates made using $I_{LOSS}$. As both $A_D$ and $L_O$ increased, $Z_{LOSS}$ estimates were dramatically higher than $I_{LOSS}$ estimates, by as much as 46%, and dissimilarity between the two strength loss estimates similarly increased for inaccurate empirical predictions of tomography measurements obtained from the hurdle

model. Although the mean bias was near zero, the maximum absolute prediction error for $A_D$ was nearly 40%, and more work is needed to examine suitable modeling approaches for decay incidence and severity. Among all oaks with decay, 17 of 135 (13%) had maximum $Z_{LOSS}$ values >70%, with $A_D$-GVB values ranging from 51%–88% (see Figure 4). These findings show that a considerable number of oaks in developed landscapes are capable of harboring high levels of internal decay in the lower trunk. Once again, arborists are better suited to providing both "liberal" ($Z_{LOSS}$-GVB) and "conservative" ($Z_{LOSS}$-VB) estimates to account for the variation.

Applying the results of strength loss formulas to trees in the field has long been a challenge, given the range of factors that contribute to tree failure [4,19]. However, there have been attempts at assigning threshold values from strength loss formulas to better guide the process of assigning risk. Coder [41] argued that when $I_{LOSS}$ exceeds 45%, trees should be deemed a hazard. Kane [47] simulated wind-induced failure in decaying *Q. rubra* by cutting voids in the lower trunk and pulling the trees to failure. He reported that no trees with $I_{LOSS}$ < 22% failed near the related decay or void during the study while all trees with $I_{LOSS}$ > 54% failed near the damage. In a post-storm analysis of various trees, including many oaks, Smiley and Fraedrich [24] recommended a range of 20%–40% strength loss as a threshold for removal, after various defects were considered. In practice, many arborists use the ratio of solid wood thickness to trunk radius ($t/R$) proposed by Mattheck et al. [48] because it's mathematically simple, but the empirically derived safety threshold ($t/R = 0.3$), a subject of considerable debate [49], lacks a rigorous theoretical justification. Moreover, the $t/R$ ratio considers failure by hollow buckling, a different failure mode than material fracture evaluated by other formulas. Also, the shell wall thickness alone is insufficient for anticipating buckling in anisotropic tissues, such as the wood in living trees, without additional information about material properties and the length of the hollow section [50].

While Ciftci et al. [26] modeled various decay scenarios and calculated the corresponding loss of moment capacity (MCL), they did not assign any threshold values due to the many variables that must be considered, most importantly the location of decay in the trunk. In accordance with Kane and Ryan [25], they cautioned against using the strength loss formulas developed by Coder and Wagener, in part because the location of decay is not considered. Burcham et al. [22] similarly demonstrated the limitations of strength loss formulas relying on simplifying assumptions about the geometry in decayed trees. In addition, both Ciftci et al. [26] and Burcham et al. [22] note that strength loss values are only valid for the sampled cross-section.

### 4.4. Sound Wood with Low Electrical Resistance

From cross-sections where no decay was detected by SoT, ERT predicted low resistance in the heartwood in nearly 90% of all cases. When maximum sonic velocities and low ER in the heartwood are found, the internal condition can be challenging to predict. Incipient decay is one explanation, where a fungal pathogen accumulates moisture in colonized wood tissue but there is not any detectable decrease in wood density [21]. Other research indicates that an increase in cation concentration significantly reduces ER in the heartwood [51,52]. Naturally high levels of cations in groundwater may be present, but it's also been speculated that fungi can mobilize cations from the soil into wood tissues during the decay process [52]. While moisture content significantly increases due to fungal colonization and decay, moisture content alone is not believed to have a significant effect on decreasing ER in the heartwood [51,52]. It was speculated that wetwood bacteria, which are known to increase cation concentrations in colonized wood tissues, were responsible for low ER in the heartwood of American elms that exhibited no decay [18]. While it is well known that elms can harbor high populations of wetwood bacteria, incidence can be locally abundant in oaks as well [8].

Temperature can also have a significant effect on ER, especially when measurements are taken near and below freezing temperatures [53]. In this study, ERT was not performed on trees at temperatures less than ~7 °C. However, after the decay process becomes more

advanced, and thus detectable by SoT, mass loss due to fungal decay will also significantly decrease ER [52]. These findings highlight that while ERT provides insights into the internal condition of sampled trees, it must be complemented with the results of SoT to properly determine whether or not internal decay is present.

### 4.5. Presence of Wood-Decay Pathogens

Overall, there was a significantly higher frequency of internal decay in the lower trunk when signs of a fungal pathogen were present. But, of the 50 trees where fruiting bodies or asexual structures of a fungal pathogen were present, five (10%) had no measurable decay in the lower trunk. These results help to reaffirm that the presence of a pathogen alone does not automatically indicate a tree is harboring decay in the lower trunk. This has been noted for several fungal decay pathogens of landscape trees [12], especially when decay is restricted to the roots [3]. Additionally, for six trees where signs of a pathogen were present at the time of sampling, two fungal genera were found co-occurring. Attack by more than one fungal decay pathogen, especially when they differ in their pattern of decay (brown rot vs. white rot), can have important implications for management.

### 5. Conclusions

Internal decay was routinely detected in the lower trunk of 186 oaks, based on 323 pairs of sonic and electrical resistance tomograms. The diameter of the sampled cross-section, the presence of symptoms, and oak species were the most important predictors of internal decay incidence. Diameter, sampling height, and oak species were the best predictors of decay severity. When intermediate sonic velocities are included, mean $A_D$ ranged from 33%–47% among the five oak species. *Quercus alba* exhibited the highest mean $A_D$ but this was only significantly different from *Q. bicolor* or *Q. palustris*, depending on tomogram interpretation. Strength loss estimates made using $Z_{LOSS}$ were 16% higher compared to estimates made using $I_{LOSS}$. More than 10% of all decaying oaks had a maximum $Z_{LOSS} > 70\%$ at the sampled cross sections. Arborists assessing decay incidence in landscape oaks should focus on symptoms of internal decay and large diameter portions of the lower trunk.

**Supplementary Materials:** The following supporting information can be downloaded at: https://www.mdpi.com/article/10.3390/f14050978/s1, Table S1: Mean area of decay ($A_D$) and mean maximum $Z_{LOSS}$ by tomography scanning height for each oak species; Figure S1: The 1909 class tree, a pin oak (*Quercus palustris*), in the Waugh Arboretum at the University of Massachusetts, Amherst; Figure S2: A black oak (*Quercus velutina*) with root and butt rot from *Niveoporofomes spraguei*; Figure S3: Overhead view showing the stump of a white oak (*Quercus alba*) that suffered root and butt rot from *Laetiporus* (white-colored fungal mycelia, brown rot and cavity in center) and *Armillaria* (white rot of cambium and sapwood in upper left); Figure S4: Overhead view showing the stump of a pin oak (*Quercus palustris*) planted in 1894 that suffered root and butt rot from *Armillaria*.

**Author Contributions:** Conceptualization, N.J.B.; methodology, N.J.B.; software, D.C.B.; formal analysis, N.J.B. and D.C.B.; investigation, N.J.B.; resources, N.J.B.; data curation, N.J.B. and D.C.B.; writing—original draft preparation, N.J.B.; writing—review and editing, N.J.B. and D.C.B.; visualization, N.J.B. and D.C.B.; project administration, N.J.B.; funding acquisition, N.J.B. All authors have read and agreed to the published version of the manuscript.

**Funding:** This work was funded by the UMass Plant Diagnostic Laboratory, University of Massachusetts, Amherst.

**Data Availability Statement:** The data presented in this study are available on request from the corresponding author.

**Acknowledgments:** The authors are grateful for the field assistance provided by Genevieve Higgins, Greg Dorr, and the many tree wardens, arborists, and homeowners that assisted with the location and access of oaks for this study.

**Conflicts of Interest:** The authors declare no conflict of interest.

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
