# Peer review of "Internal Decay in Landscape Oaks (Quercus spp.): Incidence, Severity, Explanatory Variables, and Estimates of Strength Loss"

_forests, doi:10.3390/f14050978_

Round 1

Reviewer 1 Report

How do the authors explain the different degree of tree decay  of Quercus diferent species - various chemical composition of wood, biological characteristics of fungi species, which destruct these trees or another factors?

Reviewer 3 Report

Manuscript number: forests-2317967

Title: Incidence and severity of internal decay in landscape oaks (Quercus spp.)

General comments:

It is of great significance to study the incidence and severity of trunk internal decay for forest management, and many studies have also confirmed. This study mainly focused on the comparative analysis of the incidence and severity of internal trunk decay in five landscape oak species, discussed the predictors of frequency and severity of internal decay, and computed the tree strength losses by tomograms information. The work is meaningful, and the findings will contribute to take effective conservation measures for local policy makers. However, I have some concerns about the current manuscript. Many descriptions are too cumbersome but the key points are not highlighted in Materials and Methods part, and the discussion section does not adequately discuss the key scientific issues of this study, and some of part discussions are now not relevant to the topic. Please see my following detailed comments, and I hope they will help.

Specific comments:

1)      The authors used two tomography methods to explore the incidence and severity of internal decay in oak species, but the accuracy of the two tomograms methods were not verified with the real values of decay. This is an important point that needs to be considered by the authors.

2)      L98 of Page 2 “Trees were sampled over a six-year period (2016–2021) from April into November”. It can be seen from this statement that the sampling time span was 6 years, and the degree of decay will also spread and develop over time. Moreover, different site conditions (moisture, climate, meteorological factors, etc) will also affect the accuracy of tomographic measurements. So, how did you ensure the accuracy of tomography measurement?

3)      Discussion part It was recommended that the discussion section should be discussed according to the three main research contents (proposed at the end of the introduction), such as 1) What are the advantages of the tomographic method used in this study compared to other nondestructive testing methods? --you can refer to the review article: [Soge et al., 2021, Can. J. Forest Res. 937-947] 2) What were the differences between the site conditions of oak trees and tree species in other regions? For example: [Wei et al., 2022, Forest Ecol. Manag. 120434]-(extremely arid regions), [Benítez et al., 2021.  Maderas-Cienc. Tecnol. 23:1-12]-(good moisture regions), and so on.

4)      There are few references in the past three years (2020-2022), and it is recommended to refer to the latest published research results on tree decay. 

Additional comments:

- L2 of page 1 replace “studies” with “study” Remove the “.” at the end of the title

- L30 of page 1 the “CTLA” appeared the first time in the manuscript, the full name should be given here.

- L76 of page 2 [(Burcham et al. 2019, 2023)] this format did not match the full text.

- L146-147 of page 3 [The software then produces an image with a colorimetric scale depicting wood density within the cross section.] -As far as I know, the colorimetric descripting generated by the software was not wood density, please confirm this point.

- Discussion section L53-224  a) It is missing a discussion of the results found on this article in relation with other articles. b) It is suggested that this section should be reorganized according to the main content of the manuscript.

- Based on the findings of this study, which would be the management proposal? There might be a good idea to improve the manuscript.

Round 2

Reviewer 3 Report

see attachment.
